# Swimming for Children with Disability: Experiences of Rehabilitation and Swimming Professionals in Australia

**DOI:** 10.3390/ijerph22111633

**Published:** 2025-10-27

**Authors:** Karen Graham, Katarina Ostojic, Leanne Johnston, Iain Dutia, Elizabeth Barnes-Keoghan, Georgina L. Clutterbuck

**Affiliations:** 1School of Health and Rehabilitation Sciences, The University of Queensland, St Lucia, QLD 4076, Australia; leanne.johnston@health.qld.gov.au (L.J.); iain.dutia@acu.edu.au (I.D.); g.clutterbuck@uq.edu.au (G.L.C.); 2Community Paediatrics Research Group, The University of Sydney, Camperdown, NSW 2050, Australia; katarina.ostojic@sydney.edu.au; 3Office of the Executive Director Allied Health, Children’s Health Queensland Hospital and Health Service, South Brisbane, QLD 4101, Australia; 4School of Allied Health, Australian Catholic University, Banyo, QLD 4014, Australia; 5Independent Researcher, Hobart, TAS 7000, Australia

**Keywords:** disability, children, swimming, physical activity, participation

## Abstract

Background: Swimming is a common goal for children with disability, and the most popular sport for children in Australia. This study explored swimming and rehabilitation professionals’ perceptions of swimming for Australian children with disability. Methods: Rehabilitation and swimming professionals with recent experience working with children with disability completed an online survey. Quantitative data from binary and Likert-scale questions were analysed descriptively. Qualitative data from open-ended questions was evaluated using reflexive thematic analysis and mapped to the family of Participation-Related Constructs (fPRC). Results: Ninety-one swimming and 55 rehabilitation professionals (n = 146) responded. Most were confident supporting children with disability with swimming goals (rehabilitation = 71.6%, swimming = 73.8%) but had neutral–very low knowledge of para-swimming eligibility and classification (rehabilitation = 75%, swimming = 77.7%). Ten themes (33 code groups) covering all core elements of the fPRC were identified. Barriers/facilitators included pool accessibility (physical and sensory); program availability; affordability; acceptability (of content and culture); and having accommodating professionals and programs. Professionals believed that swimming programs should develop children’s confidence, water-safety, swimming skills, and fitness. Conclusion: Rehabilitation and swimming professionals should review existing programs to ensure they meet the needs of children with disability. Further research is needed to create an action plan to improve swimming participation for Australian children with disability.

## 1. Introduction

Participation in physical activity during childhood is a key protective health factor associated with improved physical and mental health outcomes [1]. Additionally, children who are more physically active at school age are more likely to be active adults [2]. Many children with disability and their families face significant barriers to participation in physical activities and therefore participate less and in a smaller range of activities than typically developing children [3].

Swimming is the most popular sport for children in Australia [4] and remains a popular option for being physically active throughout adulthood internationally [5,6,7]. There is emerging evidence that swimming is beneficial for children with a variety of disabilities [8,9,10,11], as it provides a weight-supported environment for exercise [12] and teaches children specific water-safety skills to reduce their elevated risk of drowning [13,14].

Most published research on water-based activities focuses on hydrotherapy for children with disability [15,16,17]. While children with disability may start working on swimming and water safety skills with aquatic therapists in a hydrotherapy context [18], the focus of hydrotherapy is typically on physical therapeutic goals such as strength, postural control, or land-based functional goals. In comparison, swimming (including learn-to-swim programs, recreational swimming for physical health and fitness, and competitive swimming) provides children with life-long opportunities to participate in regular physical activity known to improve cardiorespiratory endurance, muscle strength, and gross motor skills [19,20,21]. Compared to a therapeutic environment, participation within a community setting provides opportunities for social interaction, peer-modelling [22] and increased acceptance of children with disability by their peers without disability [23].

The family of Participation-Related Constructs (fPRC) [24] provides a framework to communicate commonly reported barriers to swimming participation for children with disability, as well as potential facilitators for improved community programs. It includes factors relating to the individual (activity competence, sense-of self, and preferences) and those external to the individual that relate to their environment and context [24]. Environmental barriers to participation in community sports for children with disability include physical access or social acceptance, or personal barriers including decreased motivation or lacking confidence or knowledge of how to get involved [3,25]. Compared to land-based sports, swimming presents additional barriers to participation due to the unique aquatic environment in which it is performed [26]. Access to the physical pool environment is essential to participation, and children with disability may require increased time and support to participate compared to their typically developing peers [27,28,29]. In addition to physical access, community swimming programs may not always be equipped to safely support the unique needs of children with disability in terms of provider knowledge and skills and/or provision of individualised support or adjustments [12]. This means that research specific to aquatic environments is important to understand and address the barriers experienced by children with disability who wish to participate in swimming.

Swimming teachers represent the primary delivery agents of community-based swimming programs, while rehabilitation professionals such as physiotherapists and occupational therapists may introduce children with disability to swimming as they incorporate this activity into therapeutic interventions and physical activity goal setting. The perspectives of both professional groups regarding current practice in swimming for children with disability in Australia offer complementary viewpoints from community and clinical contexts. To our knowledge, there has not been a study that has explored these experiences. This survey study aimed to identify the current state of practice for swimming participation of children with disability in Australia, from the perspective of swimming teachers and rehabilitation professionals. It addresses three key research questions:1.What is the availability and content of swimming interventions and/or activities provided to children with disability in Australia?2.What are the facilitators and barriers to participation in community swimming activities for children with disability in Australia, as perceived by swimming teachers and rehabilitation professionals?3.What are the facilitators and barriers to providing swimming-focused interventions and/or activities to children with disability, as perceived by swimming teachers and rehabilitation professionals?

## 2. Materials and Methods

### 2.1. Design and Setting

The SPLASH I survey (Swimming Participation: Linking Australian Swimming & Health Sectors) was a cross-sectional survey study conducted between August 2023 and March 2024. This study was approved by the University of Queensland Ethics Committee (2023/HE000929).

A mixed-methods design was selected for the survey to leverage the strengths of both quantitative and qualitative approaches [30]. Quantitative questions captured measurable, generalizable data about participants’ qualifications, experience and the contexts in which swimming interventions were provided. Quantitative data served to contextualise complementary qualitative data, where open-ended questions were critical for capturing the depth and nuance of participants’ experiences relating to swimming for children with disability. By incorporating qualitative items, the survey created space for unanticipated insights to emerge and minimized the influence of preconceptions in question design.

### 2.2. Participants

Participants were recruited via purposive convenience sampling. Surveys were distributed to rehabilitation and swimming professionals via (i) governing bodies and employing organisations of rehabilitation and swimming professionals (e.g., Cerebral Palsy Alliance, AustSWIM), (ii) individual swim schools, or (iii) social media.

Organisations were identified through online directories and the research team’s professional contacts in rehabilitation and the swimming industry. Organisations distributed surveys to current staff and trainees to facilitate diversity of experience.

Inclusion Criteria: Participants were eligible if they held registration with the Australian Health Practitioner Regulation Agency (AHPRA) or professional swimming registration bodies (e.g., SWIM Coaches and Teachers Australia, AustSwim, Swimming Australia National Coaching Framework).

Exclusion Criteria: Participants were excluded if they could not answer the survey in English or had not provided relevant physical activity (rehabilitation professionals) or swimming (swimming professionals) services to children under the age of 18 years in the last 10 years. This criterion was used to ensure that responses reflected the current state of Australian swimming.

Sample Size: Power calculations were not appropriate due to the exploratory nature of this research. Sample size was determined pragmatically, aiming to capture a wide range of experiences and insights within the constraints of the study timeline and participant availability.

### 2.3. Measures

The SPLASH I survey (Appendix A) was a purpose-designed survey developed by the research team with experience in clinical rehabilitation, swimming research, and lived experience of neurodivergence.

The survey included 30–32 questions (depending on question branching) and was estimated to take participants approximately 20 min. Quantitative questions (5-point Likert scale, n = 11–14; binary yes/no questions, n = 6–7) examined qualifications, experience, knowledge and confidence with qualitative open-ended questions (n = 12) to explore the extent and nature of facilitators and barriers to swimming participation in more detail. Initial questions confirmed eligibility (n = 4) and collected demographic information relating to participants’ professional experience (n = 6–7). Questions then explored participants’ experiences of swimming activities for children with disability in Australia (n = 15–17), perceptions of the facilitators and barriers to participation in community swimming activities for children with disability in Australia (n = 3), and experiences of facilitators and barriers when providing or choosing not to provide swimming-focused interventions and/or activities to children with disability (n = 2).

### 2.4. Procedures

Participants accessed the survey anonymously through an online survey platform (Qualtrics). Potential participants who clicked on the electronic link to the research were provided with an overview of the study, completed screening and demographic questions to confirm eligibility, and provided consent prior to proceeding to the survey questions.

The survey was open from August 2023 to May 2024. Recruitment ceased when no additional responses were received for a 2-month period, indicating that further recruitment was unlikely.

### 2.5. Analysis

Quantitative data were analysed using descriptive statistics (e.g., frequency, mean, and median). When reporting numbers of responses, the following descriptors were used to communicate the extent of agreement: “almost all” >90%; “most” for 75–90%, “many” for 50–74%, “some” for 25–49% and “few” for <25% of responses [31,32].

Open-ended responses were analysed using reflexive thematic analysis [33] by two investigators from the research team (KG & GC) with significant clinical experience in aquatic physiotherapy and swimming-focused interventions for children with disability, lived experience of neurodivergence, and research experience in qualitative and quantitative research relating to sports-focused interventions (GC).

Latent codes and themes were initially independently developed, and semantic codes directly reflecting participants’ responses were grouped with related codes to form initial code groups. Code groups were developed from the data relating to activity type, and facilitators and barriers to children’s participation or professionals’ service provision. These code groups were subsequently mapped to the broader fPRC framework across its three primary domains: participation, encompassing attendance and involvement; intrinsic person-related factors including activity competence, sense of self and preferences; and environmental or contextual factors addressing accessibility, availability, accommodability, acceptability and affordability [34].

Regular meetings were held to discuss researchers’ perspectives, consider personal biases, and reach agreement regarding codes, code groups, and themes. Once code groups and themes were developed (by KG & GC), results were discussed with a third (LJ) and fourth (KO) author. Finally, the data were checked by the first author (KG) to ensure that previously classified codes were consistent with developed themes, and that the frequency of these codes was accurate where relevant. Consistent with reflexive thematic analysis principles, code frequencies were not routinely reported, as significance of themes is determined by their meaning and relevance to the research question rather than occurrence frequency [33].

## 3. Results

Of 146 eligible participants who completed the survey between August 2023 and March 2024, 91 were swimming professionals, and 55 were rehabilitation professionals. Due to the use of multiple distribution channels, the total number of professionals who received the survey invitation was not available to calculate a response rate.

### 3.1. Participant Demographics (Appendix B, Table A2)

#### 3.1.1. Swimming Professionals

Almost all (n = 86/91) swimming professionals listed their swimming qualifications from one of four organisations (AustSWIM n = 76, Swimming Coaches and Teachers Australia [SCTA] n = 7, Royal Lifesaving Australia n = 3, or Lifesaving Victoria n = 4), while few reported having degrees from higher education institutes (n = 5). Most swimming professionals had experience engaging with a person with disability outside of a swimming context, including through other work (n = 26), family members or friends with disability (n = 22), sport and community groups (n = 9), or their own lived experience of disability (n = 5). Experience teaching swimming to children with disability was equally distributed with approximately one-third of swimming professionals with 16 or more years of experience (n = 31), 6–15 years of experience (n = 27) or five or less years of experience (n = 30). Many swimming professionals (n = 55) had completed disability-specific swimming training in addition to their swimming qualification, and some had completed other disability training not related to swimming (n = 21). Swimming-specific disability training was primarily received from three organisations: AustSWIM (n = 35), Deaf Children Australia (n = 9) and Swimming Coaches and Teachers Australia (n = 3). Most training provided education about a range of disabilities (n = 43); however, impairment or diagnosis-specific training was reported for deaf/hard of hearing children (Deaf Children Australia (n = 9)), autistic children (variety of organisations (n = 8)), and children with vision impairment (Vision Victoria course (n = 1)). The level of education received regarding disability included professional development or work-based training (n = 13), University-level education (e.g., a full degree) (n = 7), diplomas with disability units (n = 5), or other disability training (n = 7).

#### 3.1.2. Rehabilitation Professionals

Of the 55 rehabilitation professionals, 53 were physiotherapists, one was an occupational therapist, and one did not specify their qualification. Rehabilitation professional participants were more experienced working with children with disability than swimming professional participants, with most having >5 years of experience (6–15 years: n = 17 and ≥16 years n = 16) and few (n = 7) having 1–5 years of experience. Many rehabilitation professionals had experience with swimming outside of a disability/rehabilitation context (n = 24) including participating in swimming (n = 17) or surf lifesaving (n = 1) themselves, providing swim coaching (n = 8) or surf coaching/volunteering (n = 2) to others, or as a parent of a para-swimmer (n = 1). Many of the participants (n = 20) had not completed any swimming education, some (n = 15) had completed swimming education for children with disability, and few (n = 5) had completed only general swimming education (i.e., swimming and water safety coaching courses without a specific disability focus).

### 3.2. Service Provision (Appendix C, Table A3)

Swimming professionals reported seeing more children per week, with most swimming professionals (77.3%) and only some rehabilitation professionals (37.5%) providing physical activity intervention or swimming activities for >15 children per week. When providing activities for children with disabilities, rehabilitation professionals were more likely to provide individual activities (rehabilitation 95%, swimming 38.6%), whereas swimming professionals were more likely to provide small-group activities (rehabilitation 5%, swimming 51.8%). Large-group (>6 participants) sessions were the least commonly reported group size (rehabilitation 2.3%, swimming professional 16.3%), although many swimming teachers (65.9%) reported commonly delivering large group sessions to typically developing children.

Swimming and rehabilitation professionals reported working with children with a diverse range of disabilities including neurodivergence, physical disabilities, intellectual disabilities, and children with sensory loss (e.g., blindness or low vision). Almost all professionals worked with autistic children (rehabilitation 92.7%, swimming 96.5%) and children with ADHD (rehabilitation 81.8%, swimming 91.8%) and most worked with children with intellectual disability (rehabilitation 87.3%, swimming 68.2%). A greater proportion of rehabilitation professionals saw children with neuromuscular (rehabilitation 69.1%, swimming 27.1%) and neurological conditions (rehabilitation 96.4%, swimming 64.7%), blindness/low vision (rehabilitation 67.3%, swimming 35.3%), and chromosomal conditions (rehabilitation 87.3%, swimming 50.6%), whereas swimming professionals reported seeing more children with limb deficiencies or amputations (rehabilitation 8.2%, swimming 52.9%) or deaf/hard-of-hearing children (rehabilitation 38.6%, swimming 60%).

### 3.3. Knowledge and Confidence

Many swimming (73.8%) and rehabilitation (69.1%) professionals reported having high or very high knowledge of supporting children with disability with swimming goals, with 25% of swimming professionals and 18.2% of rehabilitation professionals reporting very high knowledge (Figure 1a). Confidence in supporting children with swimming goals had similar distribution for both groups, with most swimming professionals (79.3%) and many rehabilitation professionals (72.7%) reporting being somewhat confident to very confident (Figure 1b). Many rehabilitation professionals (70.9%) and most swimming professionals (77.7%) reported having neutral or low knowledge of eligibility for para swimming (Figure 2a) or how classification works (Figure 2b; rehabilitation 74.1%, swimming 85.8%).

### 3.4. Qualitative Results

DTen themes were developed from 33 code groups and mapped across all three domains of the fPRC (Table 1). These are presented in Figure 3, where the relationship between the environment (represented by the pool water) and individual factors (represented by ‘splash bubbles’ formed by the person’s swimming stroke) are shown to influence attendance and involvement in swimming. The distribution of themes and codes according to specific survey question and participant group (i.e., swimming or rehabilitation professional) can be found in Appendix D.

#### 3.4.1. Environment

##### Physical and Sensory Accessibility

Many professionals considered physical and sensory accessibility to be essential to participation. Physical accessibility related to pool entry (e.g., ramps and hoists), changing facilities, pool characteristics (e.g., depth), and equipment (e.g., floatation devices). Sensory accessibility related to temperature of the pool water and surroundings, noise levels, and lighting. Many swimming and rehabilitation professionals reported that the absence of accessible facilities was a barrier which prevented children with disability participating in community swimming activities, particularly in rural areas. One swimming professional described multiple overlapping accessibility barriers: *“The environment—typically the centres are noisy, bright and overwhelming, access to changing facilities—there is often only one accessible room for older children, access into the pool—requires more support than the child has OR there is not a way for kids with severe physical disability to get into the pool.”*

##### Acceptable and Accommodating Swimming Programs

Swimming and rehabilitation professionals described a variety of acceptable and accommodating swimming programs in Australia. These included mainstream classes or centres, therapy services and supports (e.g., occupational therapy input to swimming programs), disability-specific swimming programs or clubs such as Rainbow Club [35], individual lessons, therapist-led swimming interventions (e.g., aquatic physiotherapy) and informal swimming activities (e.g., attending the pool with a support worker). Both professional groups reported that individual lessons were most likely to be able to accommodate for the different needs of children with disability, particularly when children first commenced swimming (rehabilitation 25%, swimming 27.4%). Swimming professionals were more likely to suggest mainstream group classes as an acceptable option (swimming = 41.1%, rehabilitation 25%) whereas rehabilitation professionals were more likely to suggest therapist-led swimming interventions (rehabilitation = 29.5%, swimming = 8.2%).

##### Acceptable and Accommodating Professionals

Swimming and rehabilitation professionals highlighted the importance of both individual professionals and their broader organisations being accommodating to the needs of children with disability. This included fostering an inclusive culture, being willing and able to provide individualised activities and supports, accommodating for parents and families, and collaborating with services. One swimming professional highlighted the importance of *“Understanding, lots of fun chatter, more inclusion…” and another swimming professional mentioned “Being aware of facilities that offer extra guidance and help.”* The importance of accommodating professionals was particularly highlighted during transitions, such as when starting or changing swimming activities. Regardless of the type of program, a lack of accommodating professionals and organisations was reported to be a barrier to participation in swimming for children with disability and their families.

##### Availability of Accommodating Family Supports

Both groups of professionals highlighted that supports to help families to facilitate swimming participation for children with disability would improve participation. Suggestions included education or resources to improve families’ knowledge of swimming programs and pathways in recreational or competitive swimming for children with disability, providing accommodating services that facilitate family/caregivers to participate in activities and support their swimmer in the water, and ensuring professionals are available to liaise with families and other providers.

##### Availability of Education and Resources for Professionals

Swimming and rehabilitation professionals also reported that opportunities to undertake professional development to improve their knowledge and/or skills relating to swimming for children with disability were not readily available. Participants associated the availability of training, resources and allied health support with being able to provide more accommodating supports, and their absence as a significant barrier to being able to provide acceptable services. One swimming professional described the gap in professional development pathways: *“Lack of training, time and the fact that after these kids can swim well, there’s nowhere for them to go, as our programs are really full and most just don’t fit into mainstream squads. So, it’s difficult to know what to do with them after they get to a certain skill level.”*

##### Availability of Swimming Programs

Despite many acceptable and accommodating programs and services being described, and both swimming and rehabilitation professionals demonstrating an understanding of what makes pool facilities accessible, the limited availability of programs for children with disability in pools was reported as a major barrier. Participants reported poor availability of disability-specific clubs and programs and accessible facilities, especially in non-metropolitan areas. Where programs did exist, professionals frequently reported long waitlists and fluctuating seasonal availability of pools.

##### Affordability of Programs

Affordability was commonly identified as a barrier to participation in swimming. Professionals reported that families experience several financial pressures when accessing swimming, including taking time away from paid employment to transport their child to- and from swimming, as well as costs relating to transport, pool entry, and programs. Both professional groups identified that lack of funding (e.g., through the National Disability Insurance Scheme) posed a significant barrier to swimming participation for children with disability who may require more intensive, or individualised supports, and that financial support for both providers and participants was essential to ensure that appropriate programs were available and affordable for this population.

#### 3.4.2. Individual Factors

##### Supporting Individual Needs and Preferences During Swimming Programs: Developing Activity Competence and Sense of Self

Professionals highlighted the importance of acknowledging the individual factors that may motivate or prevent a child from participating in swimming. Many swimming and rehabilitation professionals reported that activity competence goals (e.g., swimming or water-safety skills) were most common for children with disability. Goals relating to physical activity or developing sense of self through improved confidence in the water were reported less often, though one rehabilitation professional emphasized the importance of creating appropriate environments for building confidence: *“Access to a pool that is less scary—not so loud with variable depth options, so that the child can comfortably explore moving their own body through water.”* A few rehabilitation professionals also mentioned the importance of assessing physical activity competence so that activities provided are at the appropriate level for the child, while swimming professionals listed individual needs and preferences (e.g., behaviours or fear of the water) as potential barriers. When describing program content, one rehabilitation professional explained they focus on *“Water confidence and water safety skills like following instructions around water, floating, returning to the wall if they fall in towards some form of swimming stroke(s).”*

#### 3.4.3. Participation in Swimming

Few professionals mentioned participation as a swimming-related goal for children with disability. Those that did focused on attending classes, competition and other water sports. Very few elaborated on children’s experiences of involvement, with these participants listing broad goals relating to children’s enjoyment of the water, social and community engagement.

## 4. Discussion

Swimming and rehabilitation professionals who participated in the survey perceived that environmental factors are most impactful for the participation of children with disability in swimming. The availability of accessible pools and acceptable and accommodating professionals and programs, as well as the affordability of these, were considered foundational to participation. Swimming and rehabilitation professionals suggested that supportive environments that adapt to the needs and preferences of children with disability would provide opportunities for them and their families to choose to attend swimming lessons and be involved in programs, improve activity competence, support the development of their confidence in the water (sense of self), and engage in events, therefore influencing their participation. While the term participation was not explicitly reported frequently, there was a clear underlying emphasis on developing environments and supports that would improve participation outcomes. This was shown in the way participation was alluded to through social or community engagement and connections, or the transactions between individual factors and participation. Professionals’ responses in this survey demonstrated clear insight into the elements of the fPRC within their sphere of influence and control, emphasising actionable ways that they could improve environments and contexts for children with disability in swimming. The following discussion explores these themes with a focus on those which professionals highlighted as having the greatest impact if addressed in practice.

The limited availability of accessible pools reported in our survey aligns with global findings of a lack of accessible facilities restricting participation in swimming and other sports programs of children with disability [36]. Pools need to be accessible, both from a physical perspective (entry to buildings, change rooms, and supported access to pools when required) and a sensory perspective (lighting, sound, and identifying times when pools are less busy). While Australian swimming pool regulations mandate accessible pool entry (entry via ramp, zero depth entry with an aquatic wheelchair, or platform or sling-style swimming pool lift) based on characteristics of the building and pool [37,38], they do not offer further guidance on other accessibility features. For example, pool depth and temperature are both important considerations for people with disability which are not addressed. This focus on standards for safe entry without guidance on broader accessibility is common in both high-, and low- and middle-income countries [39]. Research emphasises the value of sensory accessibility guidelines [40]; however, at present, these are limited to informal guidelines published by individual advocates and organisations [41,42,43].

Despite professionals reporting high levels of knowledge and confidence in supporting children with disability in swimming programs in quantitative questions, they continued to report barriers to providing swimming activities for children with disability when asked about their experiences. Barriers included lack of experience, support, knowledge of certain disabilities or competitive sports pathways and poor availability of therapists. Swimming professionals were more likely to see children with limb differences or amputations and deaf or hard-of-hearing children, while children with more complex physical disability, such as neurological or neuromuscular conditions, were more likely to be seen by a rehabilitation professional. This is consistent with international findings that swimming teachers tend to be more comfortable teaching children with mild disability and do not feel adequately equipped to provide acceptable and accommodating supports to children with more complex disability [44,45,46]. Youth with complex or multiple disability such as neurological disability in conjunction with intellectual disability or neurodivergence require tailored multidisciplinary swimming interventions to accommodate their needs and reduce the risk of adverse events in the pool [47]. This suggests that swimming teachers and rehabilitation professionals would benefit from education and resources which improve (i) their knowledge and skills supporting children with more complex disability in swimming activities, (ii) collaboration between rehabilitation and swimming professionals in practice, and (iii) knowledge regarding para-swimming, including classification and eligibility.

Even when accommodating programs led by confident and competent professionals are available in accessible pools, they are often unaffordable for many individuals and families. Affordability includes both program and pool access costs, as well as the financial and time cost of transport [12,48]. This travel time is a significant issue in Australia and other countries with geographically dispersed populations, where access to pools in general, and particularly pools with specific accessibility features is limited. The Australian Bureau of Statistics reports that a higher percentage of people with disability live in regional areas than in city centres [49]. While the National Disability Insurance Scheme aims to improve the affordability of many disability-specific services in Australia, funding for swimming lessons is not included. Other funding sources, such as those for people with low incomes, can be difficult to access due to their variability between states, inconsistent availability, and complex application processes. Given the limited government support for swimming education access, volunteer-run programs similar to Canada’s Swimming With a Mission program [50] may offer a potential solution for children with disability in Australia. While these programs may reduce financial barriers accessing swimming, their success relies on the capacity of the organisations providing them and the availability of volunteers to support them. It is important that a comprehensive approach to ensuring affordable access to swimming programs that considers the unique barriers to access and affordability in rural areas is implemented across Australia for children with disability.

Compared to environmental considerations, children’s swimming activity competence was less frequently discussed. Professionals did, however, highlight the importance of assessing swimming and water-safety skills and setting goals for intervention, which often related to attaining specific swimming or water safety skills. Despite professionals’ desire to assess these skills for children with disability, they felt ill-prepared to do so, and were often unaware of what tools may be used to assess swimming and water safety in this population. While there are a small number of assessment tools that have demonstrated validity and reliability for measuring swimming and water safety skills for children with disability [51,52], and some assessments designed for children without disability have been adapted or modified for use with specific populations [20,53,54], there is little information available to professionals to assist with selection or guide their use in practice. Future research should include a review of the psychometric properties of tools available to measure swimming and water safety skills for children with disability with the view of developing a practical guide for professionals working with this population, for example by way of a decision tree [55,56].

### Limitations and Future Directions

This study has several strengths and limitations. The research team included physiotherapists with experience working with children with disability and supporting them in swimming. Reflexive thematic analysis methods were used to acknowledge the way in which the research team and recruitment strategy influenced the data, including that the professional backgrounds of researchers influenced the way that data were analysed and the choice of the fPRC as a guiding framework.

The survey distribution method attempted to recruit a wide range of participants from rehabilitation and swimming backgrounds. Due to the multi-faceted recruitment approach, it was not possible to calculate a precise response rate, as the number of individuals who were aware of the survey could not be determined. Individuals who had greater experience or interest in working with children with disability may have been more likely to respond to the survey, compared to the overall population of rehabilitation and swimming professionals. This may explain the reports of high confidence working with children with disability on swimming goals. The findings should be considered within the context of the absent data of professionals who did not participate due to interest, awareness or capacity.

Finally, this research reported on the perspectives of rehabilitation and swimming professionals. Future research should include meaningful connection with the disability community regarding their unique experiences and opinions in relation to swimming in Australia. This should be used alongside this paper’s findings relating to rehabilitation and swimming professionals’ experiences, and the perspectives of people with lived experience of disability that have been gathered in the United States of America and Canada [57,58]. Future research should also include the codesign of comprehensive guidelines to address the commonly reported barriers of availability and awareness of accessible facilities, and develop education and resources with the disability community to improve the knowledge and skills of professionals providing swimming interventions for children with disabilities.

## 5. Conclusions

Meaningful participation in swimming activities has the potential to increase swimming and water safety activity competence for children with disability, support the development of their sense of self by improving their confidence in the water and increase their physical activity participation. The barriers to swimming participation for children with disability identified in the SPLASH I survey underscore the need for significant change in the swimming industry to enhance participation opportunities for children with disability. Key barriers include inadequate physical and sensory accessibility, limited affordability, and poor availability of inclusive programs with accommodating professionals, particularly in rural areas. Critical steps to address these issues include improving professional education, fostering stronger connections between swimming and rehabilitation professionals, establishing clear pathways to para-swimming eligibility, expanding access to affordable, inclusive programs, and improving accessibility standards of swimming pools. Achieving these outcomes will require collaboration among swimming and rehabilitation professionals, industry stakeholders, and the disability community.

## Figures and Tables

**Figure 1 ijerph-22-01633-f001:**
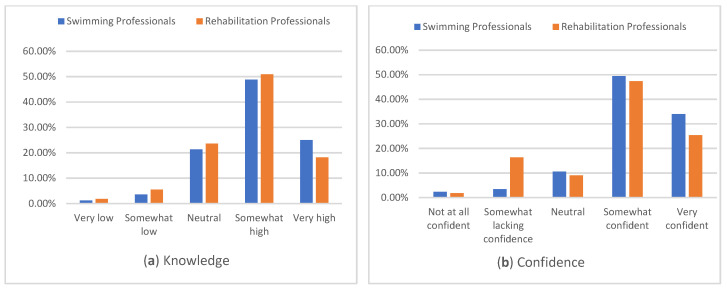
Knowledge and Confidence in Supporting Children with Disability with Swimming Goals.

**Figure 2 ijerph-22-01633-f002:**
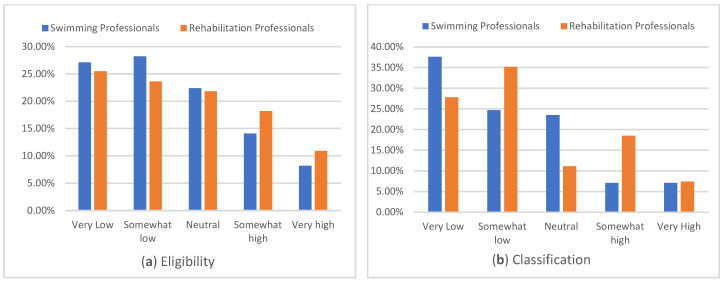
Knowledge of para-swimming classification and eligibility.

**Figure 3 ijerph-22-01633-f003:**
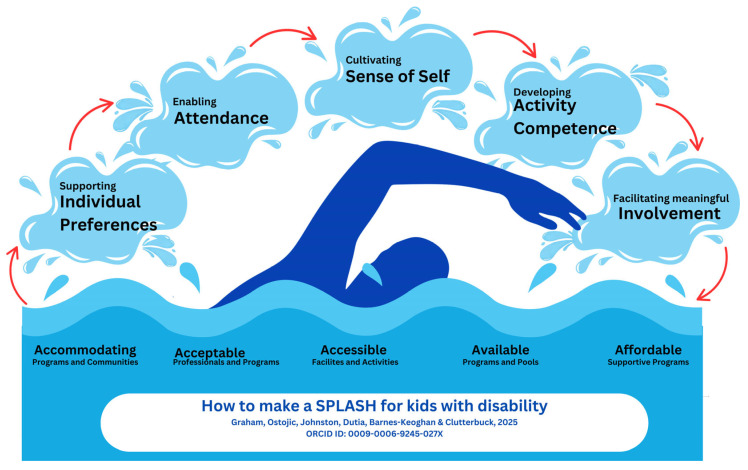
Environmental, Individual and Participation factors in swimming.

**Table 1 ijerph-22-01633-t001:** Themes and Code Groups from SPLASH survey mapped to the fPRC.

fPRC Domain	Swimming-Specific Theme	Code Groups
Environment	Accessible pools and programs	Physically accessible pools Sensory accessible pools Accessible changing facilities Accessible activities
	Affordable swimming programs	High cost of individual lessons Lack of funding (NDIS/government)
	Accommodating professionals and programs	Accommodating for service collaboration and/or therapy supports Accommodating programs Accommodating swimming teachers Accommodating for parents and families Accommodating for individualised activities and supports (including therapy supports)
	Acceptable swimming programs, activities and communities	Acceptable activities and services Acceptable disability-specific programs, school programs and mainstream programs Acceptable program content Acceptability through promoting inclusive culture
	Available swimming programs and pools	Available disability-specific swimming classes, programs and clubs Available quality programs Availability of the right opportunity to participate and progress Available (accessible) swimming pools Availability of families to engage in swimming Available professionals (who provide accommodating programs and have access to education) Available resources for families and professionals
Individual Factors	Supporting the development of Sense of Self through increasing water confidence	Development of water confidence and comfort in the water
	Supporting individual needs and preferences during swimming programs	Supporting preferences to choose and comply with swimming activities Supporting individual preferences within programs and activities
	Supporting the development of swimming and water safety skills	Assessing and developing skills -Swimming-specific skills -Water safety skills -Land-based skills -Performance-focused swimming skills
Participation	Ensuring that swimming opportunities are acceptable, accessible, available, affordable and accommodating so that children with disabilities can attend.	Quantitative data on attendance in Appendix C, Table A3 Creating opportunities to attend competitions Creating opportunities to attend other water sports
	Involvement in and enjoyment of swimming programs	Creating opportunities that enhance involvement in swimming programs and other water sports Facilitating parent and carer involvement in swimming activities

## Data Availability

Data not in the text has been made available in Appendix A, Appendix B, Appendix C and Appendix D.

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
