# Peer review of "Swimming for Children with Disability: Experiences of Rehabilitation and Swimming Professionals in Australia"

_ijerph, 2025, doi:10.3390/ijerph22111633_

Round 1
Reviewer 1 Report
Comments and Suggestions for Authors
Introduction
The research problem is unclear in the introduction. First, the research should be better problematized. It should be presented with a research question and a research hypothesis.
Materials and Methods
The method section describes the research model and explains the inclusion and exclusion criteria.
However, some shortcomings are evident.
First, how was the number of participants calculated? Was this number sufficient? A power analysis should be conducted to determine this.
Furthermore, validity and reliability information regarding the scale used is incomplete and should be added.
Results
The findings are presented in a very long and complex manner. They should be revised to be clearer, more understandable, and more concise.
Conclusion
The initial section of the discussion should include whether the research hypothesis was confirmed or not. The main findings should then be summarized and a discussion presented.
References
References should be arranged according to the spelling guide.
Author Response
Thank you for reviewing our paper. Please note that references to line numbers in the manuscript below refer to the updated manuscript.
Comment 1.
Introduction: The research problem is unclear in the introduction. First, the research should be better problematized. It should be presented with a research question and a research hypothesis.
Response 1.
Thank you for making this observation. The specific research questions that relate to the overall aim have been relocated from the methods to the introduction (Introduction, p3 L83-93).
This survey study aimed to identify the current state of practice for swimming participation of children with disability in Australia, from the perspective of swimming teachers and rehabilitation professionals. It addressed three key research questions:
- What is the availability and content of swimming interventions and/or activities provided to children with disability in Australia?
- What are the barriers and facilitators to participation in community swimming activities for children with disability in Australia, as perceived by swimming teachers and rehabilitation professionals?
- What are the barriers and facilitators to providing swimming-focused interventions and/or activities to children with disability, as perceived by swimming teachers and rehabilitation professionals?
Comment 2.
Methods: The method section describes the research model and explains the inclusion and exclusion criteria. However, some shortcomings are evident.
- First, how was the number of participants calculated? Was this number sufficient? A power analysis should be conducted to determine this.
- Furthermore, validity and reliability information regarding the scale used is incomplete and should be added.
Response 2a.
Thank you for asking for clarification regarding sample size and scale validation.
As this was an exploratory study which did not test a hypothesis, power relating to a sample size was not calculated. A justification for our method of determining sample size has been added to the Methods (2.2 Participants line 128-131).
Sample Size: Power calculations were not appropriate due to the exploratory nature of this research. Sample size was determined pragmatically, aiming to capture a wide range of experiences and insights within the constraints of the study timeline and participant availability.
Response 2b.
Regarding the survey instrument: as no validated tool exists specific to the experiences of Australian swimming and rehabilitation professionals working with children with disability in their local communities, the survey was purpose-designed by a research team with expertise in swimming interventions for children with disability and has strong face validity. Consistent with the exploratory nature of this research, open-ended questions were used to elicit deep reflection of participants’ experiences, and to obtain an understanding of the wide range of possible facilitators and barriers. There were no right or wrong answers to be scored in this survey. The Methods section 2.3 (line 133-147) outlines the survey development process and content.
Comment 3.
Results: The findings are presented in a very long and complex manner. They should be revised to be clearer, more understandable, and more concise.
Response 3.
Thank you for your suggestion regarding the presentation of the results. Text in the results has been reviewed and revised for clarity. Tables have been prioritised, and those with large volumes of quantitative data have been moved to an appendix, keeping the results section focussed on the key findings.
Comment 4.
Conclusion: The initial section of the discussion should include whether the research hypothesis was confirmed or not. The main findings should then be summarized and a discussion presented.
Response 4.
Thank you for your observations regarding the discussion. As this research was exploratory, and therefore did not have a specific hypothesis, the introductory section of the discussion (Discussion, line 388-406) has been reviewed and condensed to highlight the results in relation to the study aims (see response 1).
Swimming and rehabilitation professionals who participated in the survey perceived that environmental factors are most impactful for the participation of children with disability in swimming. The availability of accessible pools and acceptable and accommodating professionals and programs, as well as the affordability of these were considered foundational to participation. Swimming and rehabilitation professionals suggested that supportive environments that adapt to the needs and preferences of children with disability would provide opportunities for them and their families to choose to attend swimming lessons and be involved in programs, improve activity competence, support the development of their confidence in the water (sense of self), and engage in events, therefore influencing their participation. While the term participation was not explicitly reported frequently, there was a clear underlying emphasis on developing environments and supports that would improve participation outcomes. This was shown in the way participation was alluded to through social or community engagement and connections, or the transactions between individual factors and participation. Professionals’ responses in this survey demonstrated clear insight into the elements of the fPRC within their sphere of influence and control, emphasising actionable ways that they could improve environments and contexts for children with disability in swimming. The following discussion explores these themes with a focus on those which professionals highlighted as having the greatest impact if addressed in practice. The availability of accessible pools and acceptable and accommodating professionals and programs, as well as the affordability of these were considered foundational to participation.
Comment 5.
References should be arranged according to the spelling guide.
Response 5.
Thank you for reviewing the reference list. References have been numbered in order of appearance as per IJERPH’s author guidelines. We are happy to defer to the editor regarding any changes to the referencing style for publication.
Reviewer 2 Report
Comments and Suggestions for Authors
Thank you for submitting this to IJERPH.
This article is appropriate for highlighting the functions and roles of instructors for children with disabilities. It examines the qualifications of swim instructors for children with disabilities. However, I believe the design, analysis, and writing of the main text need to be re-evaluated.
Abstract
The results are inadequate. Please provide more key information, including statistical significance.
Introduction
There seem to be too many references. Most of the current information is general, and one or two references would suffice. This could be reduced slightly.
The sentence on line 46 is unnecessary here.
From the beginning of the introduction to line 51, please structure the article as follows:
"The Importance of Physical Activity for Children - Information on Swimming - Physical Activity and Limitations of Children with Disabilities"
This should be written in this manner.
Is this study about the "effects of swimming on children with disabilities"? Does it focus on "swimming instructors"? In my opinion, the purpose of the study is swim instructors, but most of the descriptions focus on swimming and children with disabilities. Therefore, the description should be appropriate to the purpose of the study. Also, break down the qualifications and abilities of swim instructors, and cite each with appropriate sentences and references.
Research Method
After reviewing this article, I offer the following suggestions:
First, divide the article into two groups.
For example,
1. Instructors with rehabilitation experience vs. without.
2. Instructors with rehabilitation training vs. without.
3. Instructors with a deep understanding of children with disabilities vs. without.
You can create other groups based on the author's ideas.
Comparing the results of their surveys, comparing them between groups using t-tests or chi-square tests, will provide more scientific and useful information.
Rather than presenting the results of a simple, horizontal survey, a better design would be to logically explain the author's stated point: the need for competencies in swim instructors for children with disabilities.
Results
Line 277 to Table 3, and line 369. Is this section appropriate for the results? There is no information regarding statistical data, proportions, or significance that can be verified numerically. This section does not seem to be the main focus. Since this study is not an observational study, measurements must be taken and the results of those measurements presented. This information regarding statistical significance applies to the entire results section of this article.
Discussion
From the beginning to line 391. This section is the results section of this study. Delete it or move it to a separate section of the discussion for comparison with other previous studies.
There is confusion as to whether this study is about children with disabilities or about coaching skills. The writing should be structured and logical, with a focus on coaches. The benefits of swimming are widely known. However, the author repeatedly mentions this in the introduction, discussion, and conclusion. This study is not about the "benefits of swimming for children with disabilities."
Conclusion
Focus on the results of this study regarding coaches and write a fact-based conclusion. It is inappropriate to write in the conclusion information that is not shown in the results of this study (e.g., lines 491-493).
Author Response
Thank you for reviewing our paper. Please note that references to line numbers in the manuscript below refer to the updated manuscript.
Comment 1.
Abstract: The results are inadequate. Please provide more key information, including statistical significance.
Response 1.
Thank you for requesting further information be included in the abstract. As this was an exploratory study, calculating statistical significance would not meet the aims of this study (See response 2 below). Rather, a qualitative analysis of the experiences of swimming and rehabilitation professionals was conducted. We provide further information in responses 4, 5 and 6 with regards to the aims of the study and statistical analysis. The abstract has been reviewed to ensure that it is under 200 words as per IJERPH’s guidelines.
Comment 2.
Introduction: There seem to be too many references. Most of the current information is general, and one or two references would suffice. This could be reduced slightly.
Response 2
The background on swimming benefits for children with disability has been reviewed and streamlined to improve flow, while ensuring the research questions remain clearly foregrounded. The first three paragraphs (physical activity importance, rationale for focusing on swimming, and swimming vs. hydrotherapy) have been condensed and streamlined, and specific research questions addressed by the survey have been relocated from the methods section to the end of the introduction for improved clarity (in response to another reviewer: Introduction, p3 L83-93).
This survey study aimed to identify the current state of practice for swimming participation of children with disability in Australia, from the perspective of swimming teachers and rehabilitation professionals. It addressed three key research questions:
- What is the availability and content of swimming interventions and/or activities provided to children with disability in Australia?
- What are the barriers and facilitators to participation in community swimming activities for children with disability in Australia, as perceived by swimming teachers and rehabilitation professionals?
- What are the barriers and facilitators to providing swimming-focused interventions and/or activities to children with disability, as perceived by swimming teachers and rehabilitation professionals?
Noting that other reviewers have requested additional references, we have carefully reviewed those included to ensure that they provide the necessary background for the current study.
Comment 3
Introduction: This could be reduced slightly. The sentence on line 46 is unnecessary here.
Response 3
Thank you for your observation. The sentence regarding Australian children starting formal swimming lessons between 0-3 years has been removed.
Comment 4
From the beginning of the introduction to line 51, please structure the article as follows:
"The Importance of Physical Activity for Children - Information on Swimming - Physical Activity and Limitations of Children with Disabilities"
Is this study about the "effects of swimming on children with disabilities"? Does it focus on "swimming instructors"? In my opinion, the purpose of the study is swim instructors, but most of the descriptions focus on swimming and children with disabilities. Therefore, the description should be appropriate to the purpose of the study. Also, break down the qualifications and abilities of swim instructors, and cite each with appropriate sentences and references.
Response 4.
Thank you for your suggestions regarding the introduction structure. We have revised the introduction to clarify the design as a descriptive survey study, moving the study aims to the end of the introduction (see response 2 above)
The justification for our focus on these professionals’ perspectives is included in the introduction, p5 L76-82:
Swimming teachers represent the primary delivery agents of community-based swimming programs, while rehabilitation professionals such as physiotherapists and occupational therapists may introduce children with disability to swimming as they incorporate this activity into therapeutic interventions and physical activity goal setting. The perspectives of both professional groups regarding current practice in swimming for children with disability in Australia offer complementary viewpoints from community and clinical contexts.
It was not possible to include details about the qualifications and abilities of all swim instructors working with children with disability in Australia as this data was not available. We have reported on the qualifications for survey participants in the results section 3.1.1 (lines 191-211).
Comment 5
Methods: First, divide the article into two groups.
For example,
1. Instructors with rehabilitation experience vs. without.
2. Instructors with rehabilitation training vs. without.
3. Instructors with a deep understanding of children with disabilities vs. without.
You can create other groups based on the author's ideas.
Comparing the results of their surveys, comparing them between groups using t-tests or chi-square tests, will provide more scientific and useful information.
Rather than presenting the results of a simple, horizontal survey, a better design would be to logically explain the author's stated point: the need for competencies in swim instructors for children with disabilities.
Since this study is not an observational study, measurements must be taken and the results of those measurements presented. This information regarding statistical significance applies to the entire results section of this article.
Response 5
Thank you for your suggestions regarding data analysis. While comparative analysis between professional groups represents an important area of future research, it was not the aim of this study.
This exploratory study aimed to describe the current practices, barriers, and facilitators for swimming participation of children with disability from professionals’ perspectives. Inferential statistics would not be appropriate for the aims of this study aim (see Response 4).
Quantitative data (Likert scales, binary questions) were analysed using descriptive statistics (frequencies, percentages, means/medians). A justification of this mixed methods approach is outlined in Methods (section 2.1, lines 101-109).
A mixed-methods design was selected for the survey to leverage the strengths of both quantitative and qualitative approaches [30]. Quantitative questions captured data about participants’ qualifications, experience and the contexts in which swimming interventions were provided. Quantitative data served to contextualise complementary qualitative data, where open ended questions were critical for capturing the depth and nuance of participants’ experiences relating to swimming for children with disability. By incorporating qualitative items, the survey created space for unanticipated insights to emerge and minimized the influence of preconceptions in question design.
Comment 6
Results: Line 277 to Table 3, and line 369. Is this section appropriate for the results? There is no information regarding statistical data, proportions, or significance that can be verified numerically. This section does not seem to be the main focus.
Response 6
Section 3.4 presents qualitative results from open-ended questions, analysed using reflexive thematic analysis as described in Methods (lines 161-183). The aim of qualitative research is not to be able to replicate the data, but to investigate meanings, experiences and perspectives [1, 2], which complement the quantitative data in this survey. Given the exploratory and meaning-generating character of the qualitative component of this research, statistical analysis would not be appropriate.
Comment 7
Discussion: From the beginning to line 391. This section is the results section of this study. Delete it or move it to a separate section of the discussion for comparison with other previous studies.
There is confusion as to whether this study is about children with disabilities or about coaching skills. The writing should be structured and logical, with a focus on coaches. The benefits of swimming are widely known. However, the author repeatedly mentions this in the introduction, discussion, and conclusion. This study is not about the "benefits of swimming for children with disabilities."
Response 7
Thank you for requesting clarification regarding the study aims. These have been clarified in the introduction (see Response 4). While the benefits of physical activity are agreed, and the benefits of hydrotherapy have also been investigated thoroughly, the specific benefits of swimming for children with disability are less well known. Since this paper will likely have a diverse audience which includes health professionals, swimming professionals, other swimming industry stakeholders and those with lived experience of disability, we believe that it is pertinent to specify these benefits. However, we have also reduced repetition where possible by refocussing the discussion (line 388-406) on the implications of professional perspectives for improving participation opportunities
Swimming and rehabilitation professionals who participated in the survey perceived that environmental factors are most impactful for the participation of children with disability in swimming. The availability of accessible pools and acceptable and accommodating professionals and programs, as well as the affordability of these were considered foundational to participation. Swimming and rehabilitation professionals suggested that supportive environments that adapt to the needs and preferences of children with disability would provide opportunities for them and their families to choose to attend swimming lessons and be involved in programs, improve activity competence, support the development of their confidence in the water (sense of self), and engage in events, therefore influencing their participation. While the term participation was not explicitly reported frequently, there was a clear underlying emphasis on developing environments and supports that would improve participation outcomes. This was shown in the way participation was alluded to through social or community engagement and connections, or the transactions between individual factors and participation. Professionals’ responses in this survey demonstrated clear insight into the elements of the fPRC within their sphere of influence and control, emphasising actionable ways that they could improve environments and contexts for children with disability in swimming. The following discussion explores these themes with a focus on those which professionals highlighted as having the greatest impact if addressed in practice. The availability of accessible pools and acceptable and accommodating professionals and programs, as well as the affordability of these were considered foundational to participation.
Comment 8
Focus on the results of this study regarding coaches and write a fact-based conclusion. It is inappropriate to write in the conclusion information that is not shown in the results of this study (e.g., lines 491-493).
Response 8
Thank you for this observation. To ensure that the conclusion focuses on the outcomes of this study, the sentence “Future research should prioritise the voices of children with disability and their families to co-design solutions that meet the diverse needs of this population.” has been removed from the conclusion and integrated into the discussion.
- Halcomb, E. and L. Hickman, Mixed Methods Research. Nursing Standard, 2015. 29(32): p. 41-47.
- Braun, V. and V. Clarke, Reflecting on reflexive thematic analysis. Qualitative Research in Sport, Exercise and Health, 2019. 11(4): p. 589-597.
Reviewer 3 Report
Comments and Suggestions for Authors
The reviewed article considers important issues of involving children with disabilities in systematic swimming classes for the purpose of health improvement.
Among the important issues considered by the author are clarification of ways for optimizing these classes through introduction of accessibility principles, increasing human resources, creating special programs, etc.
The author's vision of these issues via the prism of both swimming coaches and physical therapists’ opinion is impressive.
The author reviewed available reference sources, a significant part of which was published within the recent 5 years. The authors refer to some of their previous works, which confirms existing experience in this area.
The conducted studies correspond to a possible solution to the outlined hypothesis.
The authors present the research results using tables and figures. Instead, the use of information regarding the form of questionnaires and answers to questions significantly increases the volume of the article and burdens the data.
The conclusions correspond to the results of the study.
During the article analysis, some issues arose that can generally be taken into account for further improvement of this publication:
1. The section on research participants should describe not only the principles of forming the group of respondents, but also provide information on the number of participants, their ratio according to the field of activity, etc.
2. When presenting the results, separate subsections 3.4.1.3 and below are given from the position of presenting a generalized analysis without specifying numerical data, which, in our opinion, requires revision.
3. Subsection 3.4.1.7 Affordability of programs requires clarification and presentation of data in accordance with the possible presentation of programs as special offers for organizations, aimed at conducting classes for the specified contingent of participants, or be interpreted as the absence of methodological recommendations for swimming coaches or rehabilitation specialists in the form of classes for children with disabilities.
Author Response
Thank you for reviewing our paper. Please note that references to line numbers in the manuscript below refer to the updated manuscript.
Comment 1.
Among the important issues considered by the author are clarification of ways for optimizing these classes through introduction of accessibility principles, increasing human resources, creating special programs, etc.
The authors’ vision of these issues via the prism of both swimming coaches and physical therapists’ opinion is impressive.
The author reviewed available reference sources, a significant part of which was published within the recent 5 years. The authors refer to some of their previous works, which confirms existing experience in this area.
The conducted studies correspond to a possible solution to the outlined hypothesis.
Response 1.
Thank you for your positive comments regarding the content of the paper. We are passionate about ensuring children with disability have opportunities to participate meaningfully in swimming and are pleased to contribute to the evidence in this space.
Comment 2.
The section on research participants should describe not only the principles of forming the group of respondents, but also provide information on the number of participants, their ratio according to the field of activity, etc.
Response 2.
Thank you for seeking clarification. The placement of participant demographics in the Results section aligns with IJERPH's standard article structure and recent publications by this journal. This approach follows the principle that participant eligibility criteria (inclusion/exclusion) represent methodological design, while the actual characteristics of respondents constitute results.
For example, Barakat et al. [1] presented their survey methodology in Section 2 (including eligibility criteria), while participant characteristics were reported in Section 3.1 "Sociodemographic Characteristics" of the Results, complete with detailed demographics in tabular form.
In our manuscript, participant information is structured as follows:
- Methods section: 2.2 Participants – Recruitment (line 112-118), Eligibility including inclusion and exclusion criteria (line 119-127), sample size (line 128-131)
- Results section:
- Participants who completed the survey (Lines 185-188)
- Swimming professional participant demographics, Section 3.1.1 (Lines 190-211)
- Rehabilitation professional participant demographics, Section 3.1.2 (Lines 213-225)
This structure reflects that while we designed eligibility criteria prospectively (methodology), the actual respondents and their characteristics represent outcomes of our recruitment process (results/findings).
Comment 3.
When presenting the results, separate subsections 3.4.1.3 and below are given from the position of presenting a generalized analysis without specifying numerical data, which, in our opinion, requires revision.
Response 3.
Thank you for your comments regarding using numerical data. This was a descriptive, exploratory study which utilised multiple methods. Reflexive thematic analysis was used to analyse data from open-ended questions (described in Section 2.5, line 161-183). While some numerical data and percentages were included in 3.4.1.2 to situate the qualitative data, the focus of this section was to report on participants’ experiences and interpret meaning through the rich, qualitative data. Further justification of the mixed methods utilised within this study has been added in the
A mixed-methods design was selected for the survey to leverage the strengths of both quantitative and qualitative approaches [30]. Quantitative questions captured data about participants’ qualifications, experience and the contexts in which swimming interventions were provided. Quantitative data served to contextualise complementary qualitative data, where open ended questions were critical for capturing the depth and nuance of participants’ experiences relating to swimming for children with disability. By incorporating qualitative items, the survey created space for unanticipated insights to emerge and minimized the influence of preconceptions in question design.
Comment 4.
Subsection 3.4.1.7 Affordability of programs requires clarification and presentation of data in accordance with the possible presentation of programs as special offers for organizations, aimed at conducting classes for the specified contingent of participants, or be interpreted as the absence of methodological recommendations for swimming coaches or rehabilitation specialists in the form of classes for children with disabilities.
Response 4.
Thank you for this observation. In the results section we have presented affordability of programs as a key theme raised by participants. The broader landscape of what influences swimming program affordability, and how this may be addressed, is included in the discussion (p5, line 442-461).
- Barakat, M., et al. Perspectives of and Experience toward the Abuse of Antidiarrheal Drug (Loperamide) among Community Pharmacists: A Cross-Sectional Study. International Journal of Environmental Research and Public Health, 2023. 20, DOI: 10.3390/ijerph20146400.
Reviewer 4 Report
Comments and Suggestions for Authors
This is a valuable and timely contribution addressing the perspectives of rehabilitation and swimming professionals on swimming participation for children with disabilities in Australia. The mixed-methods approach, combined with the use of the fPRC framework, provides a solid conceptual and methodological foundation. The topic is highly relevant and novel, with practical implications for inclusive practice, professional education, and policy.
Several areas would benefit from revision before publication:
Introduction: The background is strong but overly long, with some repetition. The section would benefit from tightening and inclusion of more recent global references on inclusive sport and aquatic participation.
Methods: Adequately described, but consider reducing detail in the main text and moving some content to supplementary materials for clarity.
Results: The results are comprehensive, but some tables are overly dense and risk overwhelming readers. Consolidating key findings in the main text and relocating extended tables to the appendices would enhance readability. Qualitative results would be stronger with the inclusion of illustrative participant quotes to support themes.
Discussion: While insightful, the discussion is lengthy and occasionally repetitive. Greater integration with broader theoretical or policy frameworks would enhance interpretation. Emphasising actionable recommendations would also enhance impact.
Conclusions: Well-aligned with the findings but could be made more concise and solution-focused. Highlighting clear, practical steps for professionals, community organisations, and policymakers will increase the paper's usefulness.
Language: The English is clear overall, but sections are verbose. A light language edit is recommended for conciseness and clarity.
Comments on the Quality of English LanguageThe English is clear overall, but the text is wordy.
Author Response
Thank you for reviewing our paper. Please note that references to line numbers in the manuscript below refer to the updated manuscript.
Comment 1.
Introduction: The background is strong but overly long, with some repetition. The section would benefit from tightening and inclusion of more recent global references on inclusive sport and aquatic participation.
Response 1.
Thank you for this observation. The introduction has been revised to improve readability and reduce length. The first three paragraphs (physical activity importance, rationale for focusing on swimming, and swimming vs. hydrotherapy) have been condensed and streamlined, and key research questions have been relocated to the end of the introduction for improved clarity. (Introduction, p3 L83-93).
This survey study aimed to identify the current state of practice for swimming participation of children with disability in Australia, from the perspective of swimming teachers and rehabilitation professionals. It addressed three key research questions:
- What is the availability and content of swimming interventions and/or activities provided to children with disability in Australia?
- What are the barriers and facilitators to participation in community swimming activities for children with disability in Australia, as perceived by swimming teachers and rehabilitation professionals?
- What are the barriers and facilitators to providing swimming-focused interventions and/or activities to children with disability, as perceived by swimming teachers and rehabilitation professionals?
Regarding global references: The introduction already draws substantially on international literature on inclusive sport and aquatic participation. While the second paragraph establishes the Australian context with local statistics, and some other references are from Australian researchers, several references throughout the introduction are from international studies including Fragala-Pinkham et al’s swimming studies conducted in North America, European studies on aquatic therapy [1] and swimming [2], and a North American publication on swimming accessibility [3]. These provide a strong global perspective on inclusive aquatic participation while maintaining relevance to the Australian study context. We have reviewed the reference list to ensure it represents current international scholarship in this field.
Participation in physical activity during childhood is a key protective health factor associated with improved physical and mental health outcomes. Additionally, children who are more physically active at school age are more likely to be active adults. Many children with disability and their families face significant barriers to participation in physical activities and therefore participate less and in a smaller range of activities than typically developing children.
Swimming is the most popular sport for children in Australia and remains a popular option for being physically active throughout adulthood internationally. There is emerging evidence that swimming is beneficial for children with a variety of disabilities, as it provides a weight-supported environment for exercise and teaches children specific water-safety skills to reduce their elevated risk of drowning.
Most published research on water-based activities focusses on hydrotherapy for children with disability. While children with disability may start working on swimming and water safety skills with aquatic therapists in a hydrotherapy context, the focus of hydrotherapy is typically on physical therapeutic goals such as strength, postural control, or land-based functional goals. In comparison, swimming (including learn-to-swim programs, recreational swimming for physical health and fitness, and competitive swimming) provides children with life-long opportunities to participate in regular physical activity known to improve cardiorespiratory endurance, muscle strength, and gross motor skills. Compared to a therapeutic environment, participation within a community setting provides opportunities for social interaction, peer-modelling and increased acceptance of children with disability by their peers without disability.
Comment 2.
Methods: Adequately described, but consider reducing detail in the main text and moving some content to supplementary materials for clarity.
Response 2.
Thank you for this suggestion. We have reviewed and revised the methods section for clarity as requested. In doing so, we have retained information relevant to requests by other reviewers who had requested additional statistics and data which were not relevant to the study aims. A justification of the mixed methods approach used has been added in Methods (section 2.1, lines 101-109) as well as a paragraph regarding sample size (line 128-131). As mentioned above, the key research questions addressed by the survey have been removed from methods and inserted at the end of the introduction to clarify study aims.
Comment 3.
Results: The results are comprehensive, but some tables are overly dense and risk overwhelming readers. Consolidating key findings in the main text and relocating extended tables to the appendices would enhance readability. Qualitative results would be stronger with the inclusion of illustrative participant quotes to support themes.
Response 3.
Thank you for this feedback. Tables 1 and 2 have been revised and moved to appendices. We appreciate the request for direct quotes and have included these in sections 3.4.1.1, 3.4.1.3, 3.4.1.5, 3.4.2.1 to support key themes.
3.4.1 Environment
3.4.1.1 Physical and sensory accessibility
Many professionals considered physical and sensory accessibility to be essential to participation. Physical accessibility related to pool entry (e.g., ramps and hoists), changing facilities, pool characteristics (e.g., depth), and equipment (e.g., floatation devices). Sensory accessibility related to temperature of the pool water and surroundings, noise levels, and lighting. Many swimming and rehabilitation professionals reported that the absence of accessible facilities was a barrier which prevented children with disability participating in community swimming activities, particularly in rural areas. One swimming professional described multiple overlapping accessibility barriers: "The environment - typically the centres are noisy, bright and overwhelming, access to changing facilities - there is often only one accessible room for older children, access into the pool - requires more support than the child has OR there is not a way for kids with severe physical disability to get into the pool."
3.4.1.3 Acceptable and accommodating professionals
Swimming and rehabilitation professionals highlighted the importance of both individual professionals and their broader organisations being accommodating to the needs of children with disability. This included fostering an inclusive culture, being willing and able to provide individualised activities and supports, accommodating for parents and families, and collaborating with services. One swimming professional highlighted the importance of "Understanding, lots of fun chatter, more inclusion…” and another swimming professional mentioned “Being aware of facilities that offer extra guidance and help." The importance of accommodating professionals was particularly highlighted during transitions, such as when starting or changing swimming activities. Regardless of the type of program, a lack of accommodating professionals and organisations was reported to be a barrier to participation in swimming for children with disability and their families.
3.4.1.5 Availability of education and resources for professionals
Swimming and rehabilitation professionals also reported that opportunities to undertake professional development to improve their knowledge and/or skills relating to swimming for children with disability were not readily available. Participants associated the availability of training, resources and allied health support with being able to provide more accommodating supports, and their absence as a significant barrier to being able to provide acceptable services. One swimming professional described the gap in professional development pathways: “after these kids can swim well, there's nowhere for them to go, as our programs are really full and most just don't fit into mainstream squads. So, it's difficult to know what to do with them after they get to a certain skill level."
3.4.2.1 Supporting individual needs and preferences during swimming programs; developing activity competence and sense of self.
Professionals highlighted the importance of acknowledging the individual factors that may motivate or prevent a child from participating in swimming. Many swimming and rehabilitation professionals reported that activity competence goals (e.g., swimming or water-safety skills) were most common for children with disability. Goals relating to physical activity or developing sense of self through improved confidence in the water were reported less often, though one rehabilitation professional emphasized the importance of creating appropriate environments for building confidence: "Access to a pool that is less scary - not so loud with variable depth options, so that the child can comfortably explore moving their own body through water." A few rehabilitation professionals also mentioned the importance of assessing physical activity competence so that activities provided are at the appropriate level for the child, while swimming professionals listed individual needs and preferences (e.g. behaviours or fear of the water) as potential barriers. When describing program content, one rehabilitation professional explained they focus on "Water confidence and water safety skills like following instructions around water, floating, returning to the wall if they fall in towards some form of swimming stroke(s)."
Comment 4.
Discussion: While insightful, the discussion is lengthy and occasionally repetitive. Greater integration with broader theoretical or policy frameworks would enhance interpretation. Emphasising actionable recommendations would also enhance impact.
Response 4.
Thank you for this observation. The discussion has been revised to reduce repetition and improve flow. The opening two paragraphs have been condensed into a single paragraph that addresses how findings relate to the fPRC constructs and study aims. Subsequent paragraphs explore specific fPRC domains (environmental and individual factors) with implications for practice.
Swimming and rehabilitation professionals who participated in the survey perceived that environmental factors are most impactful for the participation of children with disability in swimming. The availability of accessible pools and acceptable and accommodating professionals and programs, as well as the affordability of these were considered foundational to participation. Swimming and rehabilitation professionals suggested that supportive environments that adapt to the needs and preferences of children with disability would provide opportunities for them and their families to choose to attend swimming lessons and be involved in programs, improve activity competence, support the development of their confidence in the water (sense of self), and engage in events, therefore influencing their participation. While the term participation was not explicitly reported frequently, there was a clear underlying emphasis on developing environments and supports that would improve participation outcomes. This was shown in the way participation was alluded to through social or community engagement and connections, or the transactions between individual factors and participation. Professionals’ responses in this survey demonstrated clear insight into the elements of the fPRC within their sphere of influence and control, emphasising actionable ways that they could improve environments and contexts for children with disability in swimming. The following discussion explores these themes with a focus on those which professionals highlighted as having the greatest impact if addressed in practice.
Comment 5.
Conclusions: Well-aligned with the findings but could be made more concise and solution-focused. Highlighting clear, practical steps for professionals, community organisations, and policymakers will increase the paper's usefulness.
Response 5.
Thank you for your observation. The final 3 paragraphs of the discussion have been put under a heading of “limitations and future directions and the conclusion has been revised to ensure that actionable steps are clear
4.1 Limitations and Future Directions
This study has several strengths and limitations. The research team included physiotherapists with experience working with children with disability and supporting them in swimming. Reflexive thematic analysis methods were used to acknowledge the way in which the research team and recruitment strategy influenced the data, including that the professional backgrounds of researchers influenced the way that data were analysed, and the choice of the fPRC as a guiding framework.
The survey distribution method attempted to recruit a wide range of participants from rehabilitation and swimming backgrounds. Due to the multi-faceted recruitment approach, it was not possible to calculate a precise response rate, as the number of individuals who were aware of the survey could not be determined. Individuals who had greater experience or interest in working with children with disability may have been more likely to respond to the survey, compared to the overall population of rehabilitation and swimming professionals. This may explain the reports of high confidence working with children with disability on swimming goals. The findings should be considered within the context of the absent data of professionals who did not participate due to interest, awareness or capacity.
Finally, this research reported on the perspectives of rehabilitation and swimming professionals. Future research should include meaningful connection with the disability community regarding their unique experiences and opinions in relation to swimming in Australia. This should be used alongside this paper’s findings relating to rehabilitation and swimming professionals’ experiences, and the perspectives of people with lived experience of disability that have been gathered in the United States of America and Canada. Future research should also include the codesign of comprehensive guidelines to address the commonly reported barriers of availability and awareness of accessible facilities and develop education and resources with the disability community to improve the knowledge and skills of professionals providing swimming interventions for children with disabilities.
- Conclusions
Meaningful participation in swimming activities has the potential to increase swimming and water safety activity competence for children with disability, support the development of their sense of self by improving their confidence in the water and increase their physical activity participation. The barriers to swimming participation for children with disability identified in the SPLASH I survey underscores the need for significant change in the swimming industry to enhance participation opportunities for children with disability. Key barriers include inadequate physical and sensory accessibility, limited affordability, and poor availability of inclusive programs with accommodating professionals, particularly in rural areas. Critical steps to address these issues include improving professional education, fostering stronger connections between swimming and rehabilitation professionals, establishing clear pathways to para-swimming eligibility, expanding access to affordable, inclusive programs, and improving accessibility standards of swimming pools. Achieving these outcomes will require collaboration among swimming and rehabilitation professionals, industry stakeholders, and the disability community.
Comment 6.
Language: The English is clear overall, but sections are verbose. A light language edit is recommended for conciseness and clarity
Response 6.
Thank you. We have carefully reviewed the paper to decrease its overall length and improve clarity.
- Vaščáková, T., M. Kudláček, and U. Barrett, Halliwick Concept of Swimming and its Influence on Motoric Competencies of Children with Severe Disabilities. European Journal of Adapted Physical Activity, 2015. 8(2): p. 44-49.
- Naczk, A., E. Gajewska, and M. Naczk, Effectiveness of swimming program in adolescents with down syndrome. International journal of environmental research and public health, 2021. 18(14): p. 7441.
- Conaster, P.J., Eric; Karabulut, Ulku, Adapted Aquatics for Children with Severe Motor Impairments. International journal of aquatic research and education, 2019.
Round 2
Reviewer 1 Report
Comments and Suggestions for Authors
Thank you for your revisions.
Reviewer 2 Report
Comments and Suggestions for Authors
I have no comments.